# Genome-Wide Identification and Characterization of the Soybean DEAD-Box Gene Family and Expression Response to Rhizobia

**DOI:** 10.3390/ijms23031120

**Published:** 2022-01-20

**Authors:** Yongliang Wang, Junwen Liao, Jun Wu, Huimei Huang, Zhanxin Yuan, Wei Yang, Xinying Wu, Xia Li

**Affiliations:** National Key Laboratory of Crop Genetic Improvement, Hubei Hongshan Laboratory, College of Plant Science and Technology, Huazhong Agricultural University, No. 1 Shizishan Road, Hongshan District, Wuhan 430070, China; wangyonglaing@webmail.hzau.edu.cn (Y.W.); liaojunwenn@163.com (J.L.); wujun6662022@163.com (J.W.); huanghuimei163241@163.com (H.H.); yuanzhanxin1218@163.com (Z.Y.); yangwei1051730475@163.com (W.Y.); 15927351341@163.com (X.W.)

**Keywords:** *Glycine max*, DEAD-box RNA helicases, phylogenetic analysis, nodulation, rhizobia

## Abstract

DEAD-box proteins are a large family of RNA helicases that play important roles in almost all cellular RNA processes in model plants. However, little is known about this family of proteins in crops such as soybean. Here, we identified 80 DEAD-box family genes in the *Glycine max* (soybean) genome. These *DEAD-box* genes were distributed on 19 chromosomes, and some genes were clustered together. The majority of DEAD-box family proteins were highly conserved in Arabidopsis and soybean, but *Glyma.08G231300* and *Glyma.14G115100* were specific to soybean. The promoters of these *DEAD-box* genes share *cis*-acting elements involved in plant responses to MeJA, salicylic acid (SA), low temperature and biotic as well as abiotic stresses; interestingly, half of the genes contain nodulation-related *cis* elements in their promoters. Microarray data analysis revealed that the *DEAD-box* genes were differentially expressed in the root and nodule. Notably, 31 genes were induced by rhizobia and/or were highly expressed in the nodule. Real-time quantitative PCR analysis validated the expression patterns of some *DEAD-box* genes, and among them, *Glyma.08G231300* and *Glyma.14G115100* were induced by rhizobia in root hair. Thus, we provide a comprehensive view of the DEAD-box family genes in soybean and highlight the crucial role of these genes in symbiotic nodulation.

## 1. Introduction

Helicases are essential enzymes found in all prokaryotes and eukaryotes and play important roles in essentially all nucleic acid metabolism processes. These helicase proteins can be divided into different families based on different conserved motifs [1]. The monomeric helicases have two RecA-like spherical domains with a linker, while hexameric helicases contain six domains arranged in a ring. The core structures of helicases also include the motifs involved in NTP binding and hydrolysis [2]. Depending on the type of target nucleic acids, helicases are classified as DNA or RNA helicases, which are involved in DNA and RNA metabolism, respectively.

The majority of RNA helicases are DExD/H-box proteins, which share conserved motifs and domain structures [3]. These RNA helicases have more than seven conserved motifs, which form two domains. Domain one is located at the N-terminal of DExD/H-box proteins and usually consists of motifs Q, I, Ia, Ib, II and III, while domain two is located at the C-terminal of these proteins and consists of motifs IV, V and VI. Both domains are interlinked by α-helices and β-folds, and the number of helices and folds varies from protein to protein [4]. Thus, the proteins form a cleft with an ATP binding site on one side and an RNA binding site on the other. This core structure of RNA helicases ensures that RNA helicases have the ability to simultaneously bind ATP and RNA to remodel RNA and RNA–protein complexes [5]. RNA helicases are classified into several subgroups, and the DEAD-box (DDX) family is the largest protein subgroup of RNA helicases among them. The DDX family contains 12 highly conserved motifs and can be easily identified by the inclusion of special DEAD (ASP-Glu-Ala-ASP) motifs. These highly conserved DDX RNA helicases are necessary for almost all aspects of the RNA metabolism process, from RNA transcription to degradation [6,7,8,9]. Recently, emerging evidence has showed that DDX helicases can act as sensors, regulators and/or effectors during viral infections [10,11], implicating the diverse roles of these DDX helicases.

In plants, *DDX* genes are involved in various biological processes of plant growth and plant response to various abiotic stresses [12]. For example, the *Arabidopsis* DDX protein low expression osmotic response gene 4 (LOS4) is required to export mRNA and modulates plant development and stress responses [13]. Several DDX proteins, AtRH3, AtRH36, AtRH7, AtRH9 and AtRH25, regulate plant growth or female gamete development through regulating RNA splicing or rRNA biogenesis in Arabidopsis [14,15], in addition to plant responses to abiotic stresses, such as cold and drought [16]. ATRH7 was also involved in ribosomal assembly [17]. DDX17-like RH30 helicase is an effective antiviral limiting factor that can block tomabusvirus replication in plants [18]. In rice, the DDX RNA helicase OsBIRH1 plays important roles in defense as well as biotic and abiotic responses [19]. OsRH42 promotes plant growth and cold tolerance through the pre-mRNA splicing pathway, and precise control of OsRH42 homeostasis is crucial for the adaption of rice to environmental temperature changes [20]. These studies suggest crucial roles of DDX RNA helicases in plant development and adaptation to changing environmental conditions, such as biotic and abiotic stresses. Studies on these model plants helped us recognize the important roles of DDX RNA helicases in plants; however, it remains unknown how these DDX RNA helicases are evolved and whether these proteins play important roles in the growth and plant–microbial interactions of crops.

Soybean (*Glycine max* [L.] Merr.) is one of most important crops worldwide. It is the largest source of protein and the second largest source (contributing about 30% of total vegetable oil) of vegetable oil [21,22]. In contrast to other staple crops, soybean is associated with nitrogen-fixing bacteria, rhizobia, and forms root nodules where bacteroides can convert atmospheric N_2_ into ammonia for plant growth [23]. Under low-soil-nitrogen conditions soybean plants produce and release flavonoid compounds into the rhizosphere to attract rhizobia to the root surface and to activate the production of signaling molecules, lipo-chitooligosaccharidic Nod factors (NFs) [24]. The perception of NFs by NF receptors in root hair cells of soybean plants leads to the activation of the nodulation signaling pathway, allowing rhizobia to infect root hairs and nodule organogenesis in the root cortex [25]. Several core nodulation signaling components have been identified, and the NMN module, which consists of GmNINa-miRNA172c-NNC1, plays a key role in soybean nodulation [26,27,28]. However, these studies have mainly focused on the transcriptional regulation of nodulation. In addition, soybean is one of most sensitive crops to abiotic stresses (e.g., drought), and rhizobial and mycorrhizal inoculation can enhance the stress tolerance as well as improve the seed quality and yield of soybean [29,30]. One can predict that there are more complex regulatory networks in soybean and that RNA helicases should have roles in modulating the post-transcriptional regulation of soybean nodulation and nitrogen fixation. Although many genes have been identified that are involved in stress response, flowering and nodulation of soybean in the past twenty years [31], there are only a couple of reports on RNA helicases in soybean. One study has shown that there are 213 putative RNA helicases in the soybean genome, which are widely distributed in chromosomes and differentially expressed during development under normal conditions [32]. Another study identified *GmRH* as the stress-responsive gene [33]. Our understanding of RNA helicases in soybean is very limited, and there are many questions that need to be answered. These include whether RNA helicases regulate soybean growth and stress responses, in particular plant symbiosis with nitrogen-fixing bacteria, rhizobia.

To gain insight into the *DDX* gene family and elucidate their roles in nodulation, we performed genome-wide identification and characterization of the subfamily of DDX RNA helicases in soybean. We identified a total of 80 *DDX* genes in the soybean genome, which are distributed on almost all chromosomes, except chromosome 12. The phylogenetic tree analyses identified 78 conserved homologs of DDX proteins in *Arabidopsis* and two DDX proteins unique to soybean. Additionally, many of these *DDX* genes were expressed in different tissues of soybean; intriguingly, some of them were responsive to nitrogen and rhizobia. Finally, real-time quantitative PCR analysis validated the expression patterns of 10 *DDX*-RNA-helicase-coding genes in response to nitrogen and rhizobia treatments. To our knowledge this is the first report of a systemic, genome-wide analysis of the *DDX* gene family and their response to symbiotic nodulation in soybean. These results provide a strong foundation for understanding the functions of the *DDX* family genes and deciphering the molecular mechanisms of RNA-mediated nitrogen responses and symbiotic nitrogen fixation.

## 2. Results

### 2.1. Identification and Characterization of the DEAD-Box Family in Glycine Max

To identify the *D**DX* genes in soybean we first searched for the DDX family members in the soybean genome using the Pfam and SMART databases, based on conserved domains of DEXDc and HELICc. A total of 258 genes were obtained and their amino acid sequences were subjected to motif analysis using MEME databases for further confirmation. Thus, we identified 80 soybean *DDX* family genes that encoded 118 DDX proteins, according to the 12 conserved motifs of the DDX family (Appendix A). We then analyzed the lengths of the protein sequences, the molecular weights and the gene lengths of all the *DDX* genes (Appendix A). The lengths of the CDS of these *DDX* genes varied between 1089 and 3606, and accordingly the lengths of the DDX family proteins ranged from 362 to 1201 amino acids with an average length of 625 amino acids.

### 2.2. The Chromosome Localization of the DEAD-Box Family

Using the TBtools software and CorelDRAW drawing software tools, the *D**DX* genes were then mapped onto the chromosomes of soybean and named with their own gene ID (Figure 1). As shown in Figure 1, the members of the *DDX* family are distributed on all the chromosomes, except chromosome 12. However, these *DDX* genes were not uniformly distributed on these chromosomes and the numbers of the *DDX* genes on these chromosomes varied, ranging from two (chromosomes 1, 4, 6 and 16) to eight (chromosomes 3 and 8) (Figure 1). Thus, the density of the *DDX* genes on 20 chromosomes ranged from 0% (chromosome 12) to 10% (chromosome 3 and 8) (Figure 1).

### 2.3. Phylogenetic and Classification Analyses of DEAD-Box Family Proteins in Soybean

In order to explore the evolution of soybean DDX proteins, we conducted a comparative study of soybean and *Arabidopsis* DDX family members. In *Arabidopsis* we identified 42 genes, which encode 57 proteins containing 12 conserved motifs (Appendix A). The evolutionary relationships between the DDX family members in *Arabidopsis* and soybean were analyzed by the maximum likelihood method (Figure 2). Phylogenetic tree analysis showed that the DDX family members of soybean and *Arabidopsis* were divided into 15 subclades (Figure 2). Among them, nearly all the soybean and *Arabidopsis* DDX proteins were closely related. Interestingly, two DDX proteins (Glyma.14G115100 and Glyma.08G231300) fell into a separate subclade, XIV (Figure 2), indicating that these DDX family proteins might have evolved new functions in soybean. To confirm the phylogenetic relationship between the DDX family members in *Arabidopsis* and soybean, we performed a synteny analysis on these DDX homologs of *Arabidopsis* and soybean. The results showed that only 39 *DDX* genes in soybean were the orthologs of 16 members in *Arabidopsis* (Appendix A).

### 2.4. Gene Structure and Conserved Motif Analyses of the DEAD-Box Family

To gain insights into the structural evolution of the DDX family in soybean we compared the conserved motifs and the compositions of introns/exons based on a new neighbor-joining (NJ) tree (Figure 3). According to their gene structures, they were divided into 13 subclades and named I to XIII (Figure 3). The structures of these soybean *DDX* genes varied greatly among the subclades, but genes within the same subclade had similar gene structures (Figure 3). The exon numbers of *DDX* gene family members varied from one to eighteen. Six genes had only one exon (7.5%), while others contained multiple exons. Interestingly, we noticed that two soybean unique *DDX* genes had the longest upstream untranscribed sequences, indicating a diverse regulation of gene transcription. Next, we analyzed the motifs in these DDX proteins on the online analysis site MEME and found that all the soybean DDX proteins contained 12 conserved motifs involved in ATP binding, ATP hydrolysis, nucleic acid binding and RNA helicase in DDX RNA helicases (Figure 3). The order of these motifs was the same in these DDX proteins, but the N-terminal and C-terminal domains, as well as the amino acid sequences between the conserved motifs of these proteins, differed greatly (Figure 3). Collectively, these results suggest that soybean DDX proteins may have typical roles in RNA metabolism, but they may have diverse functions through different targets and partners.

### 2.5. Cis Element Analysis of the DEAD-Box Gene Promoters

The transcription of a gene is regulated by the binding of transcription factors to the *cis* elements in the promoter regions [34,35]. To learn more about the transcription regulation of these soybean *DDX* genes, we analyzed the 2000 bp upstream sequences of *DDX* genes via the PlantCARE database (Figure 4A and Appendix A). In addition to the abundant amount of core promoter (TATA-box), enhancer elements (CAAT-box) and light-responsive elements, twenty-six types of *cis*-regulatory elements frequently appeared in 80 promoters. These include the *cis* elements related to plant responses to phytohormones (e.g., abscisic acid, gibberellin, auxin, jasmonic acid and salicylic acid), low temperature, pathogens, wounding, drought, circadian, meristematic activity and flavonoid biosynthesis (Figure 4A). In addition, 13 types of elements are related to development and metabolism regulation. Notably, the total number of each type of *cis* element present in *DDX* promoters varied greatly, and the *cis* elements related to jasmonic acid, abscisic acid responsiveness and anaerobic were the most common in the soybean *DDX* gene family (Figure 4A and Appendix A). Most importantly, 80 *cis* elements are related to nodule specificity in 49 *DDX* genes (Figure 4B and Appendix A). The two genes specific to soybean (*Glyma.14G115100* and *Glyma.08G231300*) contain eleven hormone-responsive elements (e.g., abscisic acid, gibberellin, jasmonic acid and salicylic acid), three drought-responsive elements, one low-temperature-responsive element, five anaerobic-inducible elements, four metabolism regulation elements (e.g., meristem, endosperm), one AT-rich DNA-binding protein element and two nodule specificity elements (Figure 4C and Appendix A). These results suggest that the expression of these *DDX* genes is regulated by phytohormones as well as biotic and abiotic stresses, and that these *DDX* genes might modulate plant responses to these internal or external stimuli.

### 2.6. Expression Profiles of the DEAD-Box Gene Family in Symbiotic Nitrogen Fixation

To investigate the possible functions of these *DDX* genes in soybean, we retrieved their expression profiles from the online transcriptome database *Glycine max* eFP Browser (http://bar.utoronto.ca/efpsoybean/cgi-bin/efpWeb.cgi, accessed on 18 November 2021) and the plant comparative genomics portal Phytozome (https://phytozome-next.jgi.doe.gov/, accessed on 12 October 2021) to analyze the expression of *DDX* genes. The expression profiles of 76 *DDX* genes in the root and nodule were mined, but the remaining four members, including the two soybean-specific *DDX* genes, were not found in the database. The expression profiles of 80 *DDX* genes were mined in root hair 12 h and 24 h after inoculation. As shown in Figure 5A, eighty *DDX* genes were differentially expressed in the root and nodule: 18.75% genes were highly expressed in the root (*Glyma.13G168000*, *Glyma.20G089600*, *Glyma.04G052900*, *Glyma.07G007400*, *Glyma.07G072700*, *Glyma.10G241900*, *Glyma.15G171800*, *Glyma.08G190100*, *Glyma.14G115100*, *Glyma.03G014700*, *Glyma.08G190100*, *Glyma.17G053200*, *Glyma.13G106200*, *Glyma.15G026100*, *Glyma.03G013000*, *Glyma.13G347900*, *Glyma.18G155900*, *Glyma.06G053200*, *Glyma.08G193700* and *Glyma.09G065600*) and the nodule (*Glyma.13G168000*, *Glyma.20G089600*, *Glyma.04G052900*, *Glyma.07G007400*, *Glyma.07G072700*, *Glyma.10G241900*, *Glyma.15G171800*, *Glyma.20G152200*, *Glyma.08G190100*, *Glyma.14G115100*, *Glyma.03G014700*, *Glyma.08G190100*, *Glyma.17G053200*, *Glyma.13G106200*, *Glyma.15G026100*, *Glyma.13G347900*, *Glyma.18G155900*, *Glyma.07G072900*, *Glyma.03G013200* and *Glyma.09G065600*), respectively, 66% of which were highly expressed in the roots and nodule. The results suggest the potential role of these genes in nodulation and symbiotic nitrogen fixation.

Next, we analyzed the expression of the *DDX* genes in root hair during rhizobial infection. The majority of these *DDX* genes were expressed at very low levels in uninfected or infected root hairs 12 h after inoculation (HAI); only seven and six genes were up-regulated and down-regulated, respectively (Figure 5B). In contrast, more than 50% of the *DDX* genes were up-regulated at 24 HAI; very few genes were down-regulated in infected root hair (Figure 5B). Interestingly, the expression levels of almost all the genes were greatly increased in uninfected root hair at 24 HAI (Figure 5B), suggesting that these genes may be involved in root hair maturation. Taken together, these data suggest that *DDX* genes play important roles in different processes of symbiotic nitrogen fixation.

### 2.7. qPCR Analysis of DDX Gene Expression in Nodules and Root Hairs

To confirm the expression patterns of *DDX* genes, we performed qPCR assays to detect the expression of 18 highly expressed *DDX* genes (Appendix A) and three mildly expressed *DDX* genes (*Glyma.13G168000*, *Glyma.03G014700* and *Glyma.20G089600*) in roots and mature nodules at 28 days after inoculation (DAI). The results showed that all the tested genes were expressed in the root and nodule, and that fifteen of them were expressed at higher levels in the root and mature nodules (Figure 6A and Appendix A). Next, we analyzed the expression of the 13 *DDX* genes which were induced by rhizobia at 24 HAI in root hair (Appendix A) and the three marker genes of nodulation initiation (*GmNIN1a*, *GmENOD40-1* and *GmENOD40-2*). The qRT-PCR results showed that three marker genes were up-regulated, only six *DDX* genes were up-regulated, one *DDX* gene was unchanged and six *DDX* genes were down-regulated (Figure 6B). The inconsistent results between microarray data and our experimental analysis potentially resulted from plant culture and rhizobial infection artifacts.

To learn more about the two soybean specific *DDX* genes we also detected the expression of *Glyma.08G231300* and *Glyma.14G115100* in mature nodules and infected root hair. The expression patterns of *Glyma.08G231300* and *Glyma.14G115100* mRNAs were similar in both mature nodules and infected root hair (Figure 6). They were expressed at low levels in root nodules, but both genes were induced by rhizobial infection at 24 HAI. These results suggest that these two *DDX* genes may be involved in common processes contributing to nodulation and symbiotic nitrogen fixation.

## 3. Discussion

The DDX family of RNA helicases is widely present in all of life [36,37]. These highly conserved RNA helicases are necessary for processes from RNA transcription to degradation metabolism, and are therefore involved in almost all aspects of living organisms [6]. In the past, studies on the DDX family in plants have mainly focused on the dicot and monocot model plants, Arabidopsis and rice [38,39]. These DDX family proteins play an indispensable role in plant development, stress adaptation and plant–microbe interactions [12]. Soybean is one of the most important crops and has an important trait, plant–rhizobia symbiosis, but the characteristics of the *DDX* gene family and their potential functions in plant growth and symbiosis, in particular, are poorly understood. In this work, we presented a comprehensive analysis of the DDX family proteins in soybean, including gene classification, chromosomal locations, phylogenetic trees and expression profiles in different tissues as well as during symbiotic nodulation. Our results provide detailed information about DDX proteins and highlight the crucial roles of DDXs in soybean–rhizobia symbiosis.

The soybean genome is a partially diploidized tetraploid [40] and has more *DDX* genes than diploid species. In this study, we identified a total of 42 and 80 DDX RNA helicases in Arabidopsis and soybean, respectively, using stringent structural criteria and updated soybean genome information (Appendix A). The marked difference in the number of DDX helicases between soybean and rice was also observed in a previous study. For example, they found 50 and 87 DDX RNA helicases in Arabidopsis and soybean, respectively [24]. However, the exact number of the DDX helicases in soybean and Arabidopsis is different between the mentioned study and ours. We speculate that this difference is mainly due to the criteria used for the identification of DDX RNA helicases. We identified the DEAD-box family members by 12 conserved motifs, while they mainly defined the DDX proteins based on the DEAD motif. The fact that the number of DDX RNA helicases almost doubled suggests that they may have evolved new functions and modulated new traits in soybean. Phylogenetic analysis reveals that 78 soybean DDX helicases have close relatives in Arabidopsis (Figure 2). Intriguingly, we found two soybean specific DDX RNA helicases with different gene and protein structures. Thus, these DDX RNA helicases may be essential not only for plant growth and environmental adaptation but also for soybean-specific traits, such as soybean–rhizobia symbiosis.

These 80 soybean *DDX* genes were separated into 13 subclades (Figure 3). The number of the DDX genes varied from two (subclades II and X) to 15 (subclade IX). These *DDX* genes contained different numbers of exons, varied lengths of introns and untranscribed regions. Interestingly, six genes in subclades III and IV contained only one exon, while others contained up to eighteen exons. Notably, the two soybean-specific genes have longer 5′ untranscribed regions than the others. The differences in gene structures may lead to different mRNA abundances and the proteins they produce. Indeed, the lengths of DDX family proteins vary greatly, ranging from 362 to 1201 amino acids. Importantly, they all contain 12 conserved ATP binding, ATP hydrolysis, nucleic acid binding and RNA helicase motifs of DDX family proteins all through the rest of the proteins, implicating the roles of these proteins in RNA metabolism. Highly divergent N-terminals, C-terminals and amino acid sequences between the conserved motifs of these DDX proteins may lead to different protein structures and functions during development and adaptation to their growth environment. 

Indeed, we found that almost all the *DDX* promoters contained *cis*-regulatory elements related to plant development and responses to the environment, including light, phytohormones (e.g., abscisic acid, jasmonic acid) and abiotic stresses (e.g., temperature, drought, etc.). It is worth noting that the numbers of *cis*-regulatory elements related to abscisic acid and jasmonic acid phytohormones are the largest (Figure 4). It is well-known that abscisic acid and jasmonic acid are responsible for plant responses to abiotic and biotic stresses, respectively [41,42]. The large number of these *cis*-regulatory elements in soybean *DDX* gene promoters suggests crucial roles of the genes in plant adaptation to changing environmental conditions. Thus, we have identified potential novel candidate genes that regulate the stress tolerance of soybean through RNA metabolism. Considering the economic importance of soybean globally and continuing challenges to soybean cultivation due to global warming, the identification of stress-related genes is of great significance in facilitating basic research of soybean stress tolerance and in accelerating breeding for stress-tolerant soybean.

Soybean is able to associate with nitrogen-fixing rhizobia to form nodules where nitrogen is fixed for plant growth [26]. Symbiotic nitrogen fixation is determined by effective rhizobial infection, nodule organogenesis and the nitrogen fixation of mature nodules [27]. One can expect that these complex morphogenic and host–rhizobia interaction processes also undergo highly active RNA metabolism. Intriguingly, we provide evidence that many *DDX* genes are potentially involved in rhizobial infection and nodule development or functionality. Firstly, the promoters of 49 *DDX* genes have nodule-related *cis*-regulatory elements (Figure 4B); secondly, more than 18% of the *DDX* genes were expressed in the root and nodule in soybean (Figure 5); and finally, many genes were responsive to rhizobia infection in root hair, especially the two soybean-specific *DDX* genes (Figure 6). Our results implicate indispensable roles of these DDX RNA helicases in the process of soybean–rhizobia interactions and symbiotic nitrogen fixation. These DDX RNA helicases are a potential resource for the elucidation of the molecular mechanisms of symbiotic nodulation and for the genetic improvement of soybean nitrogen fixation efficiency.

In summary, we systemically analyzed the DDX family genes and proteins in soybean. Our findings lay a foundation for understanding RNA dynamics and functions in soybean development and environmental adaptation. To the best of our knowledge we revealed, for the first time, the crucial roles of DDX RNA helicases in symbiotic nitrogen fixation. The future functional characterization of these symbiosis related DDX RNA helicases will provide novel insights into the post-transcriptional regulation and genetic optimization of symbiotic nitrogen fixation in soybean.

## 4. Materials and Methods

### 4.1. The Identification of the DEAD-Box Genes in Arabidopsis and Soybean

To identify all members of the DDX family members of Arabidopsis and soybean, we searched for these family members in two ways, given that all DEAD-box proteins contain two conserved domains: DEXDc and HELICc. Firstly, DEXDc and HELICc were entered into the Domain Selection dialog of the SMART website (http://smart.embl-heidelberg.de/, accessed on 16 August 2021), and then Arabidopsis and soybean were entered into the Taxonomic selection dialog. We clicked on Architecture Query to obtain the results. Secondly, we clicked on BioMart under the Tools menu of the Phytozome website (https://phytozome-next.jgi.doe.gov/, accessed on 12 October 2021), selected crops and families under the SELECT MART sub-menu and clicked on Phytozome 12 Genomes. On the page, we clicked on attributes—we clicked to specify, selected Features and checked Gene Name, Gene Start and Gene End, as well as Transcript Name. We clicked filters—clicked to specify the species name, Arabidopsis and soybean, respectively. Then, we input PF00270 and PF00271 into the PROTEIN Limit to Genes with PFAM ID and clicked Results to output the results. The sequences of all the found amino acids were input into the online conservative motifs analysis MEME (https://meme-suite.org/meme/tools/meme, accessed on 18 October 2021) to retrieve the conserved motifs. All the amino acids were analyzed one by one to confirm that they contained the 12 unique motifs of the DEAD-box family.

### 4.2. The Chromosomal Location of the DEAD-Box Genes

The GTF file of the soybean genome was downloaded from the Phytozome website. According to the GTF file and the gene IDs of the soybean DDX family members, the gene density and chromosome mapping maps were drawn by the TBtools software (College of Horticulture, South China Agricultural University, Guangzhou, China) [43].

### 4.3. Phylogenetic Trees of the DEAD-Box RNA Helicases

The amino acid sequences of all the soybean and Arabidopsis DEAD-box proteins were compared using the neighbor-joining (NJ) method of the MEGA7 program (Japan Department of Biological Sciences, Tokyo Metropolitan University, Tokyo, Japan) [44]. The images of the phylogenetic trees were drawn using MEGA7. Evolutionary tree files were imported from the Evoview online website (https://evolgenius.info//evolview-v2, accessed on 15 November 2021) on the evolutionary tree editor beautification.

### 4.4. Synteny Analysis of the DEAD-Box Genes in Soybean

The amino acid sequences of the main transcripts of *Arabidopsis* and soybean were compared with each other by the TBtools software to obtain comparison files. The synteny analysis was carried out by the MCscanX software (Department of Crop and Soil Sciences, University of Georgia, Athens, GA, USA) [45] to obtain the gene pairs of *DDX* family members with a synteny relationship between *Arabidopsis* and soybean; pictures were then drawn. 

### 4.5. Expression Analyses of the DEAD-Box Genes in Soybean

The expression data of the *DDX* genes in soybean were obtained using the online website *Glycine max* eFP Browser (http://bar.utoronto.ca/efpsoybean/cgi-bin/efpWeb.cgi, accessed on 18 November 2021). All the *DDX* gene expression data were then imported into TBtools (College of Horticulture, South China Agricultural University, Guangzhou, China) to draw the expression heat map. Next, the *DDX* genes, which are highly expressed in the roots, nodules and infected root hairs at 12 and 24 HAI were analyzed to draw an up-set Wayne map using TBtools.

### 4.6. Plant Materials and Growth Conditions

Glycine max var Williams 82 (Wm 82) was used as the plant material. Soybean seeds were germinated in sterile water for two days and then planted in solid media containing a low-nitrogen nutrient solution (25*25 disposable Petri dish). The pots were put in the plant growth room at 26 °C, with a light intensity of 200 μmol/m^2^/s, under light for 16 h and in the dark for 8 h for 3 days, allowing seed germination. The germinating seedlings were inoculated with *Bradyrhizobium diazoefficens* rhizobia strain USDA110 (OD_600_ = 0.08), suspended in sterile water. At 24 HAI the roots of inoculated and uninoculated plants were carefully removed and cleaned quickly in liquid nitrogen. We needed to quickly and gently brush the infected and uninfected root hair into a mortar. After the liquid nitrogen evaporated the samples were then quickly put into 2 mL centrifuge tubes and frozen in liquid nitrogen.

### 4.7. RNA Isolation and Real-Time Quantitative RT-PCR Expression Analysis

The total RNA of root materials was extracted using TRIpure Reagent (Aidlab Biotechnologies Co., Ltd., Beijing, China). A RT Supermix (No. R223-01, Vazyme Biotech Co., Ltd., Nanjing, China) kit was used for mRNA reverse transcription. The quantitative reagent for gene expression detection was Vazyme SYBR Green Mix (No. Q111-02, Vazyme Biotech Co., Ltd., Nanjing, China), and the quantitative PCR instrument was BIO-RAD CFX96/384 (Bio-Rad, Hercules, CA, USA). Each PCR reaction contained 10 μL 2 × STBR Green Mix, 0.5 μL of each primer, and appropriately diluted cDNA. The thermal cycling conditions were 95 °C for 5 min, followed by 40 cycles of 95 °C for 10 s and 60 °C for 30 s. The *GmELF1b* gene was used as an internal reference for all the qRT-PCR analyses. Each treatment was repeated three times independently. The primers used are described in Appendix A.

## 5. Conclusions

In this work, we identified the DEAD-box family genes and proteins in soybean in a genome-wide manner. These *DDX* genes were widely distributed in nearly all chromosomes. The majority of the DDX proteins were highly homologous with Arabidopsis DDX proteins; two soybean specific DDX proteins also had evolved. These *DDX* genes contained multiple *cis*-regulatory elements in their promoters and were differentially expressed during development and in response to rhizobia. As a sessile organism, soybean has to deal with changing environments, low-nitrogen conditions in particular; it is conceivable that a large family of DDX helicases enables them to survive in their growth environments through the modulation of RNA metabolism from biogenesis to decay. This study lay a solid foundation for a further in-depth functional analysis of the *DDX* genes, and for deciphering their implications on soybean symbiotic nitrogen fixation and a wide range of agricultural traits.

## Figures and Tables

**Figure 1 ijms-23-01120-f001:**
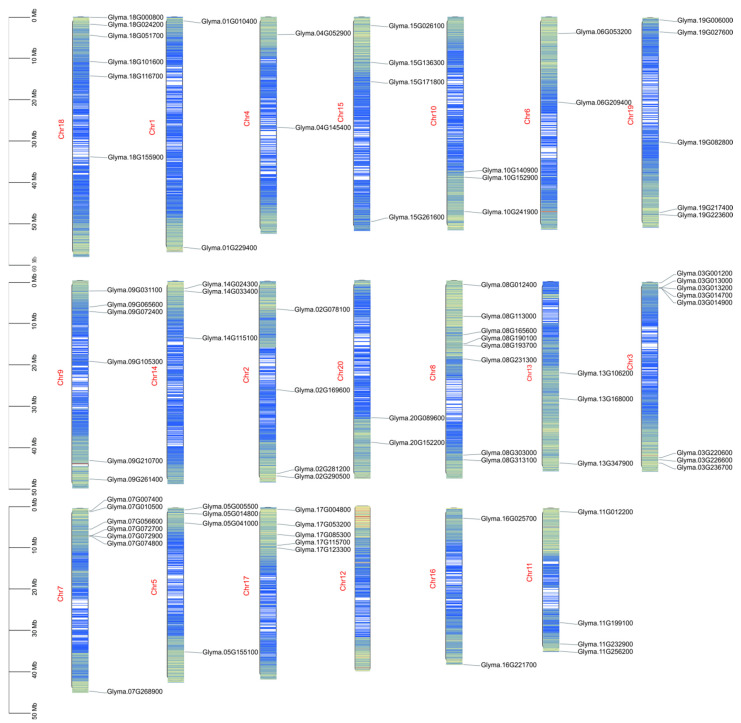
The chromosome distribution of the *DDX* genes.

**Figure 2 ijms-23-01120-f002:**
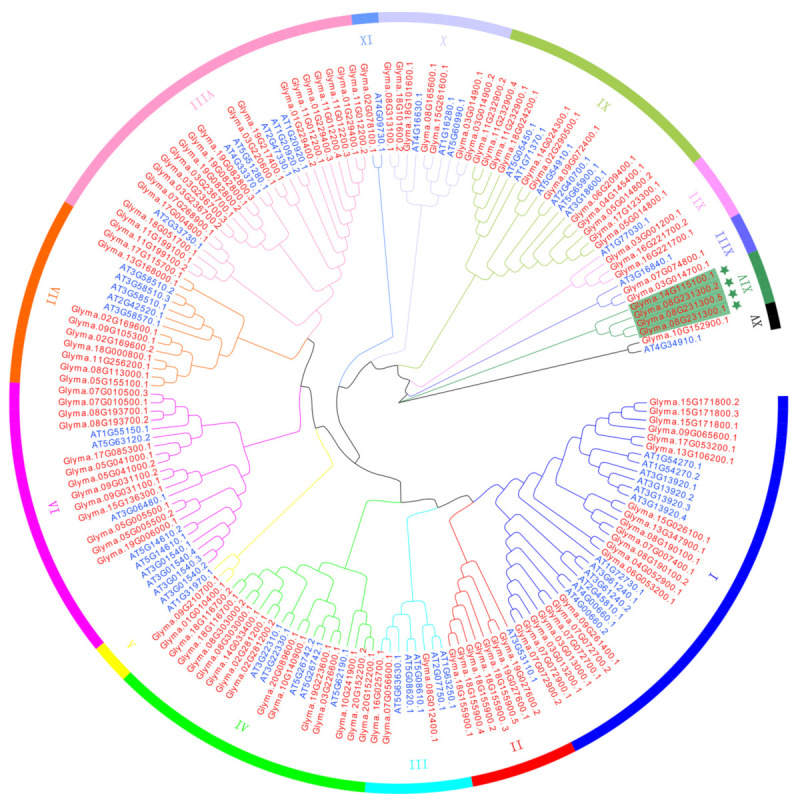
Phylogenetic tree of DDX proteins in Arabidopsis thaliana and soybean. Tree constructed with 1000 bootstrap replications. DDX proteins from Arabidopsis and soybean distinguished using blue and red, respectively. Proteins from different clusters (I–XV) were indicated with different color zones. Green stars at XIV indicate the soybean-specific DDX proteins.

**Figure 3 ijms-23-01120-f003:**
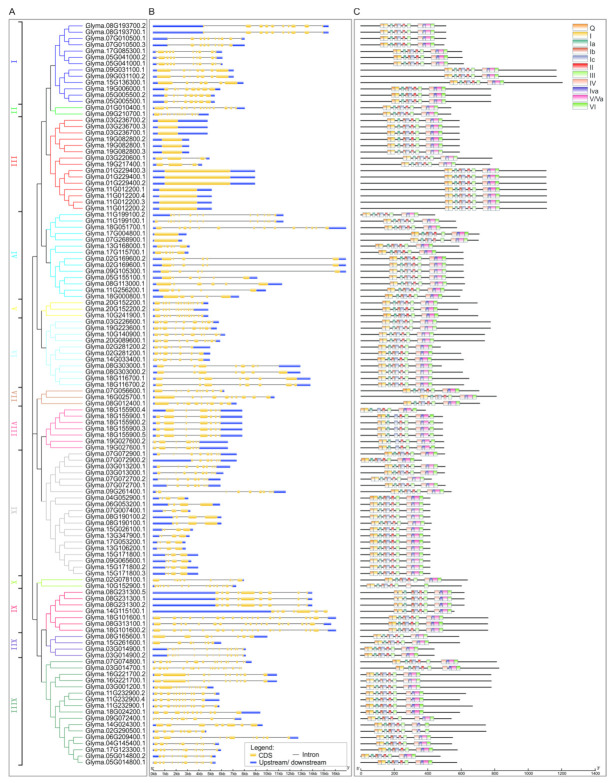
The structures and motif compositions of *DDX* genes and proteins. (**A**) Phylogenetic tree and classification of DDX proteins was constructed by MEGA. (**B**) Gene structures of *DDX* genes. The yellow boxes, black lines and blue boxes represent the CDS, intron and upstream/downstream. (**C**) Twelve conserved motif compositions of DDX proteins were identified using MEME. Each color represents a specific motif.

**Figure 4 ijms-23-01120-f004:**
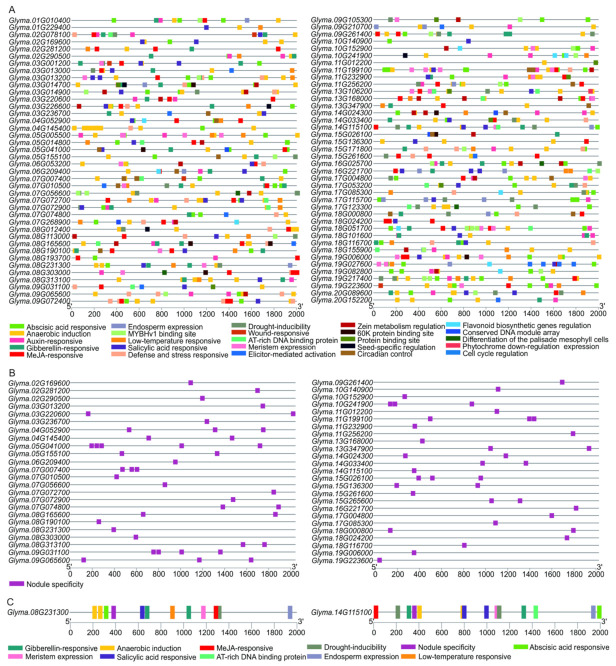
Predicted *cis* elements in the promoter regions of *DDX* genes. (**A**) Predicted *cis* elements of all *DDX* genes. (**B**) Predicted nodule-specific *cis* elements of 49 *DDX* genes. (**C**) Predicted *cis* elements of two soybean specific *DDX* genes. The scale bars at the bottom indicate the lengths of the promoter sequences. Color boxes indicate *cis* elements in the promoter regions.

**Figure 5 ijms-23-01120-f005:**
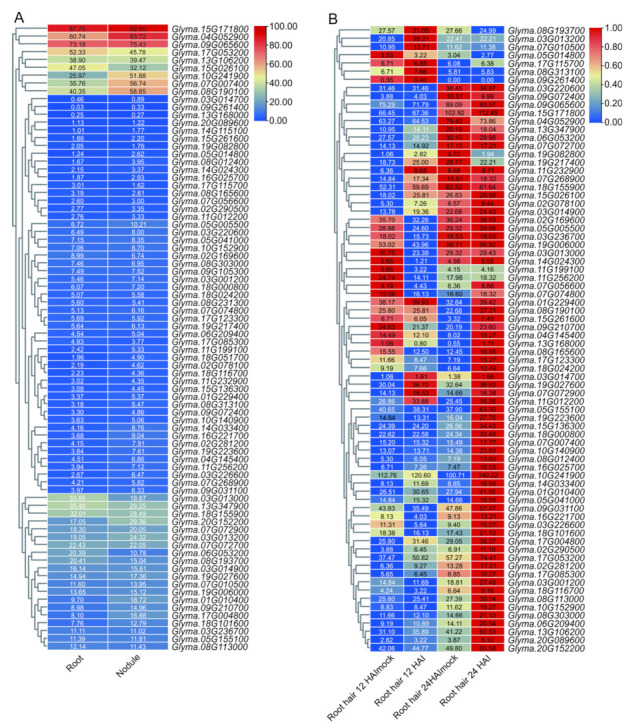
Expression analysis of *DDX* genes in the root and nodule, and their response to rhizobium. A heat map with clustering was created based on the RPKM value of 80 *DDX* genes of soybean. (**A**) Expression patterns of *DDX* genes in the root and nodule. (**B**) Expression patterns of *DDX* genes’ response to rhizobium in root hair at 12 and 24 h after inoculation (HAI). The colored scale varies from green to red, which indicates low or high levels of gene expression. Root hair 12 HAImock/Root hair 24 HAImock and Root hair 12 HAI/Root hair 24 HAI indicate root hairs at 12 h/24 h after inoculation without and with rhizobia, respectively.

**Figure 6 ijms-23-01120-f006:**
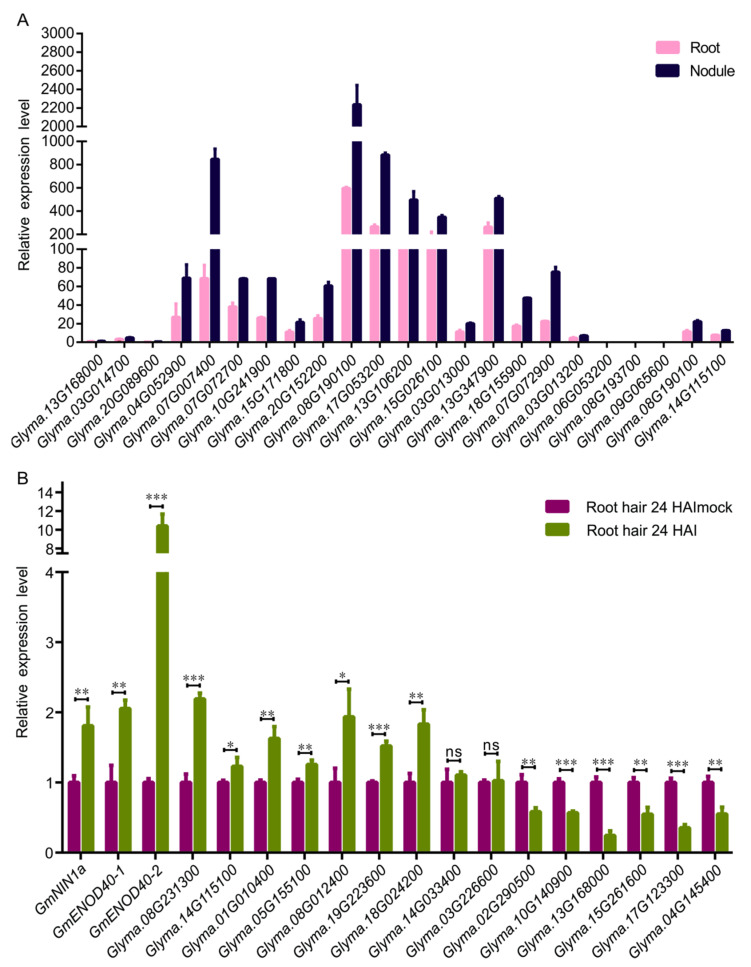
Expression profiles of *DDX* genes in the root and nodule, and their response to rhizobium. (**A**) Expression patterns of selected *DDX* genes in the root and nodule were examined by a qPCR assay. (**B**) Expression patterns of selected *DDX* genes in response to rhizobium in root hair were examined by a qPCR assay at 24 HAI. Data were the most representative of three biological replicates. The *GmELF1b* gene was used as an internal control. Root hair 24 HAImock and Root hair 24 HAI indicate root hairs at 24 h after inoculation without and with rhizobia, respectively. A Student’s *t*-test tested the significance of the difference between two groups. * *p* < 0.05. ** *p* < 0.01. *** *p* < 0.001.

## Data Availability

The datasets used and/or analyzed during the current study are available from the corresponding author on reasonable request. However, most of the data is shown in Appendix A.

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
