# Peer review of "Genome-Wide Identification and Characterization of the Soybean DEAD-Box Gene Family and Expression Response to Rhizobia"

_ijms, 2022, doi:10.3390/ijms23031120_

Round 1
Reviewer 1 Report
In the current manuscript, the authors have provided a genome-wide inventory of DEAD-Box genes in soybean. The authors have identified 80 DEAD-Box genes in the soybean genome. Further, chromosomal localization, gene, and protein architecture were studied followed by phylogenetic analysis. Various cis-regulatory elements including hormone-regulated and nodule-specific elements were identified in the promoter regions of these genes. In silico expression profile of these genes showed an upregulated expression primarily in root hair that led to the final expression study through qRT-PCR suggesting an induced expression pattern under rhizobium infection. Overall, the study has significant scientific importance and the authors have provided satisfactory evidence to support further studies. However, some minor revisions are required.
The manuscript needs attention in writing. Check carefully for typographical and grammatical mistakes. External language professional help could be a significant help.
The images are of low resolution. Replace these images with high resolutions. Figures 1,3,4 need special attention and figure 2 and 5 can be improved.
Synteny analysis and interaction network (Co-expression analysis) would be a significant addition to the manuscript. Consider the addition of these studies.
Check the citations of figures and tables in the manuscript carefully.
Statistical analysis in figure 6 is required.
Expand the abbreviations mentioned in the manuscript (e.g., HAI). In qRT-PCR what does “RH-R 24h” signify, mention the expanded form in the manuscript.
Why are the authors mentioning ‘root hair’ in qRT-PCR? RNA was extracted from whole roots, if not authors need to mention that RNA was extracted from root hair and only then mentioning ‘root hair’ in qRT-PCR have a significance. Otherwise, more qRT-PCR with root-specific marker genes and root hair-specific marker genes would be required.
Author Response
Response to Reviewers’ Comments
Responses to reviewer 1 comments
Overall comments: In the current manuscript, the authors have provided a genome-wide inventory of DEAD-Box genes in soybean. The authors have identified 80 DEAD-Box genes in the soybean genome. Further, chromosomal localization, gene, and protein architecture were studied followed by phylogenetic analysis. Various cis-regulatory elements including hormone-regulated and nodule-specific elements were identified in the promoter regions of these genes. In silico expression profile of these genes showed an upregulated expression primarily in root hair that led to the final expression study through qRT-PCR suggesting an induced expression pattern under rhizobium infection. Overall, the study has significant scientific importance and the authors have provided satisfactory evidence to support further studies. However, some minor revisions are required.
Our response: We are deeply grateful to you for your precious time and constructive suggestions that help us to improve our manuscript.
Comment 1: The manuscript needs attention in writing. Check carefully for typographical and grammatical mistakes. External language professional help could be a significant help.
Response 1: Thank you for the good suggestions! According to your suggestion, we have carefully optimized the layout of the article and invited professional English scholars to optimize the language of the manuscript.
Comment 2: The images are of low resolution. Replace these images with high resolutions. Figures 1,3,4 need special attention and figure 2 and 5 can be improved.
Response 2: Thank you for your critical suggestion! We have improved the resolution of all images and replaced the original images with higher quality images.
Comment 3: Synteny analysis and interaction network (Co-expression analysis) would be a significant addition to the manuscript. Consider the addition of these studies.
Response 3: Thank you for your constructive suggestion! As you see, we have included the synterny analysis results and co-expression analysis in the revised manuscript (Supplemental Figure S2andS4). These will provide more information for the relationship between the genes and possible network of the genes that are involved in.
Comment 4: Check the citations of figures and tables in the manuscript carefully.
Response 4: Thank you for your strict suggestion! We carefully checked all references to ensure that each reference appeared in the correct position in the manuscript.
Comment 5: Statistical analysis in figure 6 is required.
Response 5: Thank you for your critical suggestion! We have added statistical difference analysis in Figure 6.
Comment 6: Expand the abbreviations mentioned in the manuscript (e.g., HAI). In qRT-PCR what does “RH-R 24h”signify, mention the expanded form in the manuscript.
Response 6: Thank you for your constructive suggestion! We have added expanded form to some acronyms. For instance, HAI is hours after rhizobial inoculation, RH-R 24h and RH+R 24h replace with Root hair 24 HAImock and Root hair 24 HAI. Root hair 24 HAImock and Root hair 24 HAI indicate root hairs at 24 hours after inoculation without and with rhizobia, respectively, etc.
Comment 7: Why are the authors mentioning‘root hair’in qRT-PCR? RNA was extracted from whole roots, if not authors need to mention that RNA was extracted from root hair and only then mentioning ‘root hair’ in qRT-PCR have a significance. Otherwise, more qRT-PCR with root-specific marker genes and root hair-specific marker genes would be required.
Response 7: Thank you for your important suggestion! In this study, the samples used for qRT-PCR were extracted from root hair. We also described the extraction process in the material method. We have also described it in the appropriate place in our manuscript to avoid misunderstanding of the results.
Reviewer 2 Report
This is an interesting study and the authors have collected a unique dataset using cutting edge methodology. The paper is well written and structured. The science structure and overall theme of the manuscript are sound and acceptable. In my opinion, the overall concept is interesting and important. The paper is well written and is a worthy contribution that will be of interest to the readers of the IJMS journal.
The authors provide a comprehensive view of the DEAD-box family genes in soybean and highlight the crucial role of these genes in symbiotic modulation and support the idea of the research of DEAD-box proteins in soybean.
The manuscript is interesting, associated with actual plant science trends. The presentation of the paper in the form of graphs and tables is clear and the analyses are well described.
All results are comprehensible. In general, the paper is clear and well-written. The quality of the manuscript is overall good considering the methodological approach, reliability of the results, and the ability of authors to discuss all the data properly. Include the introduction of the missing information (research gaps). Why it is required to run such specific research? What are the alternative solutions?
Some arguments need clearer and tighter presentation, more understandable for a large spectrum of plant biologists.
The paper brings new aspects and novelties. The discussion could be improved to make the manuscript easy to follow, and more attractive and critical. Read/ use papers:
https://doi.org/10.1186/s12870-021-02949-z; https://doi.org/10.1111/pbi.13682; https://doi.org/10.1111/ppl.13454; https://doi.org/10.1371/journal.pone.0056312
Results are applicable in applied research and practice and can provoke new experimental activities. The paper can also provide a basis for a better understanding of the impacts and mitigation mechanisms of climate change on crop production. The paper brings new original aspects and the novelty of the paper is OK. The manuscript needs further careful revision and then it could be accepted for publication. MINOR changes are needed in order to have a high-quality manuscript.
Author Response
Response to reviewer 2 comments
Overall comments: This is an interesting study and the authors have collected a unique dataset using cutting edge methodology. The paper is well written and structured. The science structure and overall theme of the manuscript are sound and acceptable. In my opinion, the overall concept is interesting and important. The paper is well written and is a worthy contribution that will be of interest to the readers of the IJMS journal.
The authors provide a comprehensive view of the DEAD-box family genes in soybean and highlight the crucial role of these genes in symbiotic modulation and support the idea of the research of DEAD-box proteins in soybean.
The manuscript is interesting, associated with actual plant science trends. The presentation of the paper in the form of graphs and tables is clear and the analyses are well described. All results are comprehensible. In general, the paper is clear and well-written. The quality of the manuscript is overall good considering the methodological approach, reliability of the results, and the ability of authors to discuss all the data properly. Include the introduction of the missing information (research gaps). Why it is required to run such specific research? What are the alternative solutions? Some arguments need clearer and tighter presentation, more understandable for a large spectrum of plant biologists.
Our response: We are deeply grateful to you for your precious time and constructive suggestions that help us to improve our manuscript. According to your suggestion, we have included the introduction for the reason to conduct this research, the alternative solutions, and have made it clearer and more understandable for plant biologists.
Comment 1: The paper brings new aspects and novelties. The discussion could be improved to make the manuscript easy to follow, and more attractive and critical. Read/ use papers:
https://doi.org/10.1186/s12870-021-02949-z;https://doi.org/10.1111/pbi.13682; https://doi.org/10.1111/ppl.13454; https://doi.org/10.1371/journal.pone.0056312
Response 1: Thank you for your encouragement and good suggestions! In particular, we thank you for giving us an example paper, so we can revise the discussion.
Comment 2: Results are applicable in applied research and practice and can provoke new experimental activities. The paper can also provide a basis for a better understanding of the impacts and mitigation mechanisms of climate change on crop production. The paper brings new original aspects and the novelty of the paper is OK. The manuscript needs further careful revision and then it could be accepted for publication. MINOR changes are needed in order to have a high-quality manuscript.
Response 2: Thank you for your critical suggestion! We have made extensive revision throughout the manuscript to improve the quality of the manuscript. These include revisions of introduction and discussion, improvement resolution of the figures, copy editing of the manuscript, and so on. I believe that with these revisions, the manuscript has been greatly improved.